# Modified Epoxy Resin on the Burning Behavior and Mechanical Properties of Aramid Fiber Composite

**DOI:** 10.3390/ma17164028

**Published:** 2024-08-13

**Authors:** Xuke Lan, Chenxi Bian, Yunxian Yang, Qi Zhang, Guangyan Huang

**Affiliations:** 1National Key Laboratory of Explosion Science and Safety Protection, Beijing Institute of Technology, Beijing 100081, China; lanxuke@bit.edu.cn (X.L.);; 2Beijing Institute of Technology Chongqing Innovation Center, Chongqing 401120, China; 3Beijing Institute of Technology Zhuhai, Zhuhai 519088, China; 4Advanced Research Institute of Multidisciplinary Sciences, Beijing Institute of Technology, Beijing 100081, China

**Keywords:** aramid fiber/epoxy resin composite, flame retardancy, mechanical properties, ballistic impact, flame-retardant mechanism

## Abstract

Aramid fiber/epoxy resin (AF/EP) composite has been heavily used as an impact protection material due to its excellent mechanical properties and lightweight merits. Meanwhile, it is also necessary to concern the flammability of matrix resin and the wick effect of aramid fiber, which would constitute a fire risk in harsh environments. In this work, a multifunctional flame-retardant modifier (EAD) was incorporated into the AF/EP system to improve the flame retardation. The addition of 5 wt% EAD made the AF/EP composite exhibit a high limiting oxygen index (LOI) value of 37.5%, self-extinguishment, as well as decreased total heat release and total smoke release. The results from thermogravimetric analysis (TGA) and dynamic mechanical analysis (DMA) demonstrated that the treated composites maintained good thermal stability. Due to the combined action of covalent and noncovalent bonds in the matrix-rich region, the interfacial bonding improved, which endowed AF/EP composite with strengthening and toughening effects. Compared with the control sample AF/EP, the tensile strength and ballistic parameter (V_50_) of the sample with 5 wt% EAD increased by 17% and 10%, accompanied with ductile failure mode. Furthermore, the flame-retardant mechanism was obtained by analyzing the actions in condensed and gaseous phases. Thanks to good compatibility and interfacial adhesion, the incorporation of EAD solved the inconsistent issue between flame retardancy and mechanical properties, which further expanded the application of AF/EP composite in the protection field.

## 1. Introduction

High-performance fiber-reinforced polymer composites with superior mechanical properties have been developed and utilized in many high-tech fields, involving aerospace, military equipment, automobile, and so on [1,2,3]. Amongst the major fibers, aramid fiber (AF) is considered to play an important role in advanced ballistic protection materials due to its low density, high specific strength and modulus, as well as excellent impact toughness [4,5,6,7]. In the aramid fiber-reinforced composite system, matrix resin, maintaining the integrity of fiber and distributing the load during high-speed impact, can also affect the ballistic performance through delamination, debonding behavior, and back face deformation [8,9]. Thanks to its significant adhesion and outstanding mechanical strength, epoxy resin (EP) is widely served as a matrix to prepare the aramid fiber-reinforced composite [10,11,12].

However, epoxy resin exhibits inherent flammability and their cured composites also have the issue of high fire risk due to the wick effect, in which heat and fuel are produced and transferred along the fiber direction to continuously feed the fire [13,14,15]. Therefore, it is necessary to improve the flame retardancy of the aramid fiber-reinforced epoxy resin composite (AF/EP), allowing for its further application in harsh environments. The main strategy to endow high-performance fiber-reinforced polymer composites with flame retardation is to incorporate diverse functional ingredients into matrix resin [16,17], and meanwhile, phosphorus-containing compounds have become promising flame retardants because of their high efficiency, good designability, and low toxicity [18,19,20,21].

Take a carbon fiber-reinforced composite for example. Flame retardation was enhanced via introducing a phosphorous-containing transesterification modifier named DPEA into epoxy resin [22]. Because of the covalent bonding between DPEA and epoxy, the composite possessed intrinsic flame retardancy, achieving an increased LOI value and UL-94 V-0 rating. Another phosphorus-derived imidazolium (CEPM-1) was used as a curing agent for epoxy, and the wick effect of the corresponding carbon fiber-reinforced composite was inhibited, resulting in an LOI value of 42.5% and V-0 classification in UL-94 test [13]. Other phosphorus-containing flame retardants were also added into epoxy resin to fabricate carbon fiber-reinforced composite, of which the flammability was suppressed significantly [23,24]. Additionally, the flexural strength and interlaminar shear strength of the treated composites were also improved, which was attributed to enhanced interfacial adhesion [25]. Some significant results have been achieved using flame-retardant epoxy resin and its carbon fiber-reinforced composites. However, there are few studies that focus on the flame-retardant AF/EP system according to the currently available literature. As a protection material, fire safety is as essential as ballistic performance for the AF/EP composite.

Generally, the flame retardancy of the composites is always improved at the expense of the deterioration of mechanical properties, and the aramid fiber presents an obvious inert surface and water sensitivity [26,27]. Therefore, it is still a great challenge to tackle these issues and meet the requirement for practical application. Tailoring the interfacial performance of composites is an effective way to prepare the AF/EP composite with outstanding performance [28,29,30]. In this study, a phosphorus-containing flame retardant (EAD), which was designed and prepared previously in our lab, was used as multifunctional modifier to solve the mismatch of flame retardancy and mechanical properties. How to preserve good mechanical and ballistic performance was a key point via interfacial adhesion. The effect of EAD on the thermal behaviors, flame retardancy, mechanical properties, and ballistic performance of AF/EP was investigated systematically. Moreover, the flame-retardant mechanism was confirmed by analyzing the actions in condensed and gaseous phases.

## 2. Experiment

### 2.1. Materials

The aramid fabric (PF-J09, plain-weave) with an areal density of 200 g/m^2^ and thickness of 0.33 mm was kindly supplied by Hunan Air Defense Technology Co., Ltd., Changsha, China. The thread count of the warp and weft yarns is 106 yarns/10 cm, and the tensile strengths of warp and weft yarns are 9090 N/5 cm and 10,010 N/5 cm, respectively. Epoxy resin (DGEBA: epoxy value of 0.51 mol/100 g) was provided by Guangzhou Huixin Chemical Co., Ltd., Guangzhou, China, and the curing agent (DDM: 4,4-diamino-diphenylmethane) was purchased from Shanghai Macklin Biochemical Co., Ltd., Shanghai, China. The flame-retardant EAD was prepared according to the previous method [31].

### 2.2. Preparation of AF/EP Composite

The AF/EP composites were fabricated via hand lay-up and hot-press processes. Firstly, AF was soaked in ethanol solution for 30 min under ultrasonic conditions, and then dried at 60 °C for 30 min. Then, the well-mixed resin with 9 wt% ethanol as a diluent was applied to impregnate the AF. Afterwards, the infiltrated AF/EP prepregs were put into the hot presser as well as degassed under the pressure of 1 MPa 5 times, and then the cured laminates were obtained after the procedure with 3 MPa at 100 °C for 2 h and 150 °C for 2 h. Finally, the specimens with different dimensions were cut using a high-velocity water jet machine (JJ-II42-2020, Jinjian, Shanghai, China).

According to the previous results, the percentage of flame retardant was set as 2 wt% and 5 wt% in total amount of epoxy resin, which were named as sample AF/EP/2%EAD and AF/EP/5%EAD, respectively. In addition, the control sample AF/EP without EAD was also prepared under the same conditions, and the matrix content in all the samples was kept at 32 ± 1 wt%. The whole fabrication process is illustrated in Figure 1a.

### 2.3. Characterization

Thermogravimetric analysis (TGA) and dynamic mechanical analysis (DMA) were used to study the thermal performance of the AF/EP composite. Each sample with 20 ± 0.5 mg was heated from 30 to 700 °C at the rate of 10 °C/min under a nitrogen flow of 90.0 mL/ min on the equipment (TGA550, TA Instruments, New Castle, DA, USA). Specimens with 60.0 mm × 10.5 mm × 3.0 mm were tested in three-point bending configuration with the constant frequency of 10 Hz on an analyzer (TAQ800, TA Instruments, New Castle, DA, USA), and the temperature was set from 30 to 180 °C at the heating rate of 5 °C/min.

A limiting oxygen index (LOI) (HC-2C, Shangyuan, Nanjing, China) test, UL-94 vertical burning (CZF-4, Shangyuan, Nanjing, China) test, and cone calorimeter test (CCT: FTT, West Sussex, UK) were carried out to investigate the burning behavior of AF/EP composites. Correspondingly, the dimensions of the samples were set at 127.0 mm × 6.5 mm × 2.0 mm, 127.0 mm × 13.0 mm × 2.0 mm, and 100.0 mm × 100.0 mm × 3.0 mm according to standard ASTM D 2863-2017, ASTM D 3801-2019, and ISO 5660-2015, respectively. Additionally, the heat flux of CCT was set at 50 kW/m^2^.

In order to investigate the flame-retardant mechanism, functions in condensed and gas phases were analyzed by using relative methods including scanning electron microscopy (SEM), Raman spectroscopy, pyrolysis combustion flow calorimeter (PCFC), and thermogravimetric analysis coupled with Fourier transform infrared (TG-IR). PCFC was performed at the range of 100–900 °C on an apparatus (FTT Fire Testing Technology Ltd., West Sussex, UK) according to ASTM D 7309-07. TG-IR was used on the TGA550 instrument coupled with an FTIR spectrometer (Thermo Fisher, Waltham, MA, USA). The procedure for TGA was kept the same as the previous one, and the gaseous products were recorded at the wavenumbers of 500–4000 cm^−1^, simultaneously. SEM was conducted under vacuum environment at a voltage of 15 kV on an equipment ProX (Thermo Fisher, USA), and the Raman was performed at room temperature with an excitation wavelength of 532 nm on LabRam HR (Horiba, Kyoto, Japan).

Mechanical properties of AF/EP laminates were studied via tensile and interlaminar shear strength (ILSS) tests on a universal testing machine (MTS E45, MTS Systems, Chicago, IL, USA). Samples (250.0 mm × 25.0 mm × 3.0 mm) were prepared for tensile test at a crosshead speed of 2 mm/min following the standard ASTM 3039-2017. In accordance with the ASTM D 2344-2016 standard, the ILSS of specimens with 25.0 mm × 6.0 mm × 3.0 mm was measured in a short-beam shear mode with span length of 12 mm and loading speed of 1 mm/min. At least 5 specimens were loaded for each test. Aiming to analyze the interfacial adhesion of AF/EP laminates, the fracture surface of laminates was observed using the same SEM equipment.

The ballistic performance of AF/EP laminates was studied via a high-speed impact test with a barrel of 2500 mm × 20 mm. As shown in Figure 1b, the samples were fixed between two steel plates with a 100 mm diameter target region, while the projectile, made of steel ball with a mass of 2.1 g and 8 mm diameter, was also propelled to the end of the barrel through a PVC pipe. To observe the response process of the target and measure the impact velocity of the composite projectile, two high speed cameras were arranged, one parallel to the ballistic line and the other set at 45 degrees to the back of the target. Both of the two high speed cameras were Photron FASTCAM UX100 (Photron, San Diego, CA, USA), the frame rate of which was 25,000 fps. The ballistic limit velocity (V_50_) of AF/EP laminate was tested according to NATO STANAG 2920, which was calculated by averaging six values. At least 10 samples were tested to obtain six correlative speeds, including three lowest complete penetration speeds and three highest partial penetration speeds. During impacting, the energy absorption was calculated under the premise that the lost kinetic energy of the projectile was fully absorbed by the AF/EP laminates, and which, therefore, can be expressed by the Equation (1).
(1)$$\Delta E=1/2\;m\;(V_{i}^{2}-V_{r}^{2})$$
where the ΔE was the energy adsorption, m (kg) was the mass of projectile, V_i_ (m/s) and V_r_ (m/s) were the initial velocity and residual velocity, respectively.

## 3. Result and Discussion

### 3.1. Thermal Properties of AF/EP Composite

#### 3.1.1. Thermogravimetric Analysis

TGA was firstly conducted to evaluate the effect of EAD on the thermal degradation behavior of the AF/EP laminate. The results are presented in Figure 2a,b and Appendix A. Due to the excellent thermal stability, there was almost no mass loss for pure aramid fiber before 500 °C. Only one degradation peak appeared at 560 °C, caused by the random breakage and decomposition of the molecular chain, along with the residue of 39.8% at 700 °C. Meanwhile, the maximum degradation rate of epoxy resin was detected at 377 °C, derived from the pyrolysis of its benzene ring structure, N-O, and ether bonds. And 18.5% residue was left at 700 °C. With regard to the AF/EP composites, two obvious peaks were observed at around 377 °C and 560 °C, which were attributed to the degradation of matrix and fiber, respectively. In comparison with sample AF/EP, the addition of EAD barely changed the T_max1_ and T_max2_ of the composites, but the corresponding maximum degradation rates reduced with the content of flame retardant. Furthermore, the initial degradation temperature (T_5wt_%) and residue were also affected by the incorporation of EAD. Especially for sample AF/EP/5%EAD, the T_5wt_% decreased at 359 °C and the residue at 700 °C rose up to 39.7%. This is because of the premature decomposition of EAD to promote the carbonization of the matrix, and the charring formation can protect inner material from further decomposition. In addition, the mass loss of the AF/EP composites was approximately 30% between 350 °C and 500 °C, and this indicated the mass content of the matrix as well as the uniformity of these composites [32,33].

#### 3.1.2. Dynamic Mechanical Analysis

From the dynamic mechanical analysis (DMA), parameters relating to the viscoelastic behavior of the AF/EP laminate were observed along with the elevating temperature. The storage modulus (E′) can be used to assess the elastic structure of material to resist the loads, and Tan δ, calculated from the ratio of E″ to E′, demonstrates the damping performance. The E′, Tan δ, and glass transition temperature (T_g_) of AF/EP composites are shown in Figure 2c,d. Compared with sample AF/EP, the composite with 2 wt% EAD presented a higher E′ value and the one with 5 wt% EAD showed a lower value before the T_g_. The improvement can be explained by the increased stiffness of epoxy resin, while some defects in the fiber/matrix interface leads to a negative effect, resulting in a decrease at the 5 wt% loading [34]. The variation in tan δ, relating to molecular movement and interfacial fraction at the fiber/matrix interface, also confirmed the negative effect of overloaded EAD on the interfacial bonding between fiber and matrix [35]. Concerning the T_g_ from Appendix A, the addition of EAD led to a preservation of the thermal stability of the AF/EP composite, although there was a slight decrease in the modified samples. This is attributed to the change in the crosslinking density of the matrix resin, resulting from the combining action of chemical reaction, crosslinking points, and steric hindrance between EAD and EP. Overall, introducing the flame retardant can influence the interfacial region and maintain the excellent dynamic response of the AF/EP composite.

### 3.2. Burning Behavior of AF/EP Composite

#### 3.2.1. Flame Retardancy

Firstly, LOI and UL-94 tests were carried out to study the effect of EAD on the burning behavior of the AF/EP composite. The corresponding results are listed in Appendix A. The reference sample AF/EP showed an LOI value of 31.5% and burned out during the UL-94 test. After introducing EAD into the system, the LOI value increased with the content of flame retardant, reflecting as 35.0% for AF/EP/2%EAD and 37.5% for AF/EP/5%EAD. During the UL-94 test, samples with flame retardant exhibited self-extinguishment, and the average afterflame time reduced to 36 s for sample with 5 wt% EAD. The digital photos of samples during UL-94 test are presented in Figure 3a. As for the untreated sample, the wick effect of aramid fiber and flammability of epoxy resin led to a strong burning behavior. The incorporation of EAD improved the flame retardancy, while the efficiency in AF/EP composite was still inferior to the one in epoxy resin due to the wick effect [14,22].

The flame retardancy of AF/EP laminates was further evaluated through the inspection of data from CCT, which were usually used to simulate the real fire situation. The critical results are presented in Table 1 and Figure 3b. Due to the earlier decomposition of the flame retardant, the time to ignition (TTI) values of samples with EAD were lower than that of sample AF/EP. The heat release and smoke release of AF/EP composites were suppressed after incorporating EAD, which was reflected by the reduced heat release rate (HRR), peak heat release rate (PHRR), total smoke release (TSR), and total smoke production (TSP). Moreover, the maximum average rate of heat emission (MARHE), which was considered as an effective and comprehensive parameter to assess the heat release performance [36], also decreased from 152 kW/m^2^ in AF/EP to 134 kW/m^2^ in AF/EP/5%EAD. Compared with the untreated sample, a little decrease in average effective heat combustion (av-EHC) and increased residue were found in the treated samples. Combining the residue and av-EHC, which was linked with combustion efficiency [37], it can indicate that the EAD played the main role in the condensed phase.

#### 3.2.2. Flame Retardant Mechanism

First of all, SEM and Raman were used to analyze the flame-retardant action of EAD in the condensed phase. The difference in residue morphology of AF/EP composites can be seen in Figure 4a,b. As for sample AF/EP, an external surface with big holes and split yarns was observed, and the internal surface showed yarns with little char because the matrix burnt out during combustion. In comparison, samples with EAD exhibited a more compact external layer with well-protected yarns, while the internal surfaces were covered with massive chars, displaying a more continuous layer in sample AF/EP/5%EAD. Furthermore, the D band at 1362 cm^−1^ and G band at 1583 cm^−1^ in the Raman spectra were used to describe the amorphous carbon structure and ordered carbon structure, respectively [38]. The ratio of ID and IG was calculated to evaluate the graphitization of the residue, showing in Figure 4c. The presence of EAD gradually decreased the value from 3.74 of sample AF/EP to 3.09 of sample AF/EP/5%EAD, which suggested the formation of a good protective layer. Consequently, the compact char layer can effectively obstruct the exchange of heat and mass during combustion.

Moreover, the flame-retardant influence of EAD in the gas phase was also examined via PCFC and TG-IR. PCFC was used to assess the complete combustion of fuel gases from the decomposition process of AF/EP composites under inert atmosphere [39]. From Figure 5a, several important parameters can be seen, including heat release rate (HRR), heat release capacity (HRC), and total heat release (THR). All the samples demonstrated two heat release peaks at around 400 °C and 600 °C, which were on account of the decomposition of the epoxy matrix and aramid fiber, respectively. The peak HRR values gradually decreased after incorporating EAD, especially for the peak at 400 °C. Analogic tendencies were also observed in HRC and THR, and this indicated that the flame-retardant action of EAD also functioned in the gas phase by inhibiting the generation of combustible gases [40]. In Figure 5b, two absorption peaks, resulting from the predominant degradation of epoxy resin and aramid fiber, were found during the heating process of both untreated and treated samples. After adding EAD into the AF/EP system, the release of total gas products reduced, and the time when the peaks appeared was about 0.3 s earlier. This is because the existence of EAD facilitates the process of dehydration and carbonization to form a barrier layer, which is consistent with the data from TGA. Concerning the categories of gas products, Figure 5c illustrates the FTIR spectra of volatiles from the AF/EP composites at the two absorption peaks. Similar products were detected in both samples, which consisted of O-H/N-H bond absorption (3600–3900 cm^−1^), hydrocarbons (3000 cm^−1^), CO_2_ (2350 cm^−1^), carbonyl (1715 cm^−1^), aromatic and aliphatic compounds (1510 cm^−1^ and 720 cm^−1^, respectively), and C-O-containing products (1177–1262 cm^−1^) [41]. This insinuates that the introduction of EAD hardly affects the decomposition route of an AF/EP cured system.

Based on the comprehensive analysis of its actions in the condensed and gas phases, the flame-retardant mechanism of EAD can be explained as follows. On the one hand, under catalyzed carbonization, a charring layer with a phosphates structure was formed to act as a good physical obstruction for heat and mass. On the other hand, EAD pyrolyzed and produced PO^•^/PO_2_^•^ free radicals to quench the HO^•^/H^•^ free radicals derived from the matrix, which can effectively reduce the fuel release in the gas phase. Accompanying the quenching effect, the generation of some noncombustible volatiles (NH_3_, H_2_O) can also dilute the concentration of combustible gases. Overall, the combination of condensed and gas functions accounted for the improvement in the flame retardancy of the AF/EP composite.

### 3.3. Mechanical Properties of AF/EP Composite

#### 3.3.1. Tensile Properties

Depending on the properties of the reinforcement, matrix, and interfacial adhesion, the tensile properties of AF/EP composites, shown in Figure 6a, were obtained to assess the influence of EAD on the in-plane properties of these laminates [42]. Compared with the tensile strength of sample AF/EP (458 MPa), treated samples demonstrated higher values with increasing EAD loading, resulting in 520 MPa for AF/EP/2%EAD and 536 MPa for AF/EP/5%EAD. As for the Young’s modulus, all the samples showed the same value. This is because the modulus is an inherent property of composites, which is dominated by the fiber’s property in composites. The addition of a small amount of flame retardant has a limited effect on the modulus. Moreover, the front and profile photos of AF/EP composites after failure are also presented in Figure 6b. Remarkable delamination and an intact fabric surface were observed in the sample AF/EP. After adding EAD into the system, the delamination was improved, and meanwhile, obvious fiber damage could be found on the front surface. This is an implication that the flame retardant can enhance the bonding strength of the matrix-enriched region based on the combined action of covalent and noncovalent bonds between the matrix resin and EAD. The improvement is beneficial for transferring stress and dissipating energy. Finally, the failure mode varied from the dominant delamination to multiple combination, involving fiber fracture, matrix crack, and delamination [43].

#### 3.3.2. Interlaminar Shear Property

Interlaminar shear strength (ILSS) was obtained via a short beam shear test, which can reflect the matrix toughness and adhesion strength between the fiber and matrix [29]. Figure 6c presents the ILSS of AF/EP laminates, and the values exhibit a fluctuation with the increase in EAD content. Compared with sample AF/EP (17.7 MPa), the value increased to 18.3 MPa for AF/EP/2%EAD and then decreased to 16.9 MPa for AF/EP/5%EAD. This is because the flame retardant was incorporated via epoxy matrix instead of aramid fiber, and some defects were probably produced in the fiber/matrix interface. When the content of EAD rose up to 5 wt%, the defects started dominating the ILSS, although the adhesion strength improved in the matrix-rich region [44]. As for the load-displacement curves in Figure 6d, akin failure modes were observed after the maximum loading. Sample AF/EP exhibited a peak load of 430 N and a decrease before failure. Relative to the untreated composite, samples with EAD also maintained a steady load after the peak value. The ductile response behavior is attributed to the toughening effect of EAD in the matrix-rich region and the inelastic deformation of aramid fiber [45]. In summary, the appropriate addition of EAD can keep the interlaminar shear property of the AF/EP composite at the same level.

The strengthening and toughening effect of AF/EP composites was investigated by SEM. In Figure 7, the micromorphology of these laminates after tensile test illustrates different fracture behavior. Due to its poor adhesion, there was little matrix on the fiber surface and obvious gaps between yarns in the control sample AF/EP, which was a typical delamination failure. In contrast, the treated samples demonstrated rough and complex micromorphology. Take sample AF/EP/5%EAD for example, in which a significantly fractured matrix and split fiber were observed, as well as many deflected cracks induced by EAD particles. The difference in the fracture morphology confirms that the incorporation of flame retardant can optimize the failure mode due to the enhanced interfacial adhesion and good dispersion in the matrix-enriched region [46], which is dominated by covalent and non-covalent bonding between matrix and EAD.

### 3.4. Ballistic Performance of AF/EP Composite

As a well-known protection material, it is essential to evaluate the effect of EAD on the ballistic performance of AF/EP composites, including V_50_, areal density, specific-V_50_, energy absorption (EA), specific energy absorption (SEA), and back face deformation (BFD). The corresponding parameters, shown in Table 2, were obtained via high-speed impact coupled with a high-speed photography method. Compared with untreated sample AF/EP, the V_50_ increased by 21.7 m/s and 33.6 m/s for AF/EP/2%EAD and AF/EP/5%EAD, respectively, while the areal density of composites rose slightly with the content of EAD. The specific V_50_, a reliable parameter calculated by the ratio of V_50_ and areal density [47], gave a first upward and then downward trend. However, all the AF/EP laminates still preserved the same level. 

Concerning the EA and SEA, which were obtained from the Equation (2), the introduction of EAD led to a gradual increase for both parameters. This phenomenon further confirmed that the presence of EAD enhanced the capacity of energy dissipation. When the projectile impacts the target, the AF/EP composites can absorb kinetic energy through a complex process, involving fiber break, deformation of back face, tension of primary yarns, deformation of secondary yarns, matrix crack and delamination, and so on [4,48]. Considering the same conditions in fiber properties and preparing process, the improved ballistic performance can be attributed to the increased interfacial adhesion in the matrix-rich region, which consumed more energy via cracks deflection and fracture [49,50].
(2)$$\mathrm{SEA}=(1/2\;mV_{2}^{50})/(\mathrm{Areal}\;\mathrm{density})$$

With regard to the BFD, all the samples demonstrated a cone shape on back panel, illustrated in Figure 8. At the same initial velocity of 370 m/s, sample AF/EP was perforated, while the treated samples were unpenetrated. In comparison with sample AF/EP/2%EAD, a lower deformation height was observed in the sample with 5 wt% EAD, which could have resulted from the moderate increment of the matrix stiffness. Overall, the incorporation of EAD brought out an improvement in ballistic performance via a positive effect on the matrix properties.

## 4. Conclusions

Aiming to improve the flame retardancy of the AF/EP composite, a multifunctional flame-retardant EAD was incorporated into the system. Under the addition of 5 wt% EAD, the AF/EP composite demonstrated good thermal stability as compared to the untreated sample. The values of T_5wt%_ and T_g_ slightly decreased by 10 °C and 6 °C, respectively. Due to the combination of a physical barrier, quenching effect, and dilution behavior, the sample with 5 wt% EAD achieved a high LOI value of 37.5% and a self-extinguishment with afterflame time of 36 s. On the contrary, the untreated AF/EP composite had an LOI value of 31.5% and burned out with severe combustion behavior. The introduction of EAD also enhanced the mechanical properties of the AF/EP composite, yielding increased tensile strength of up to 17% and good ductile failure mode at the 5 wt% loading. The strengthening and toughening effects were attributed to a good interfacial adhesion in the matrix-rich region. Furthermore, the improvement in the matrix was also beneficial to ballistic performance. Compared to the control sample, treated composite with 5 wt% EAD presented higher V_50_ and SEA values. This study gave a proposal to obtain an AF/EP composite with good comprehensive properties, which would further expand its application in the protection field. In summary, this study improved the flame retardancy, mechanical properties, and ballistic performance of the AF/EP composite. Future research efforts can be focused on improving the moisture absorption and UV resistance of the aramid composites to enhance their overall performance.

## Figures and Tables

**Figure 1 materials-17-04028-f001:**
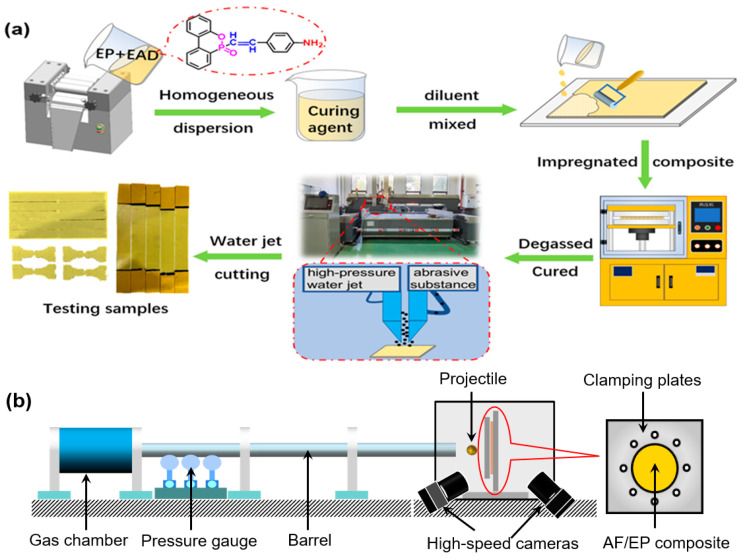
Diagram of preparation process of AF/EP laminates (**a**) and ballistic impact test (**b**).

**Figure 2 materials-17-04028-f002:**
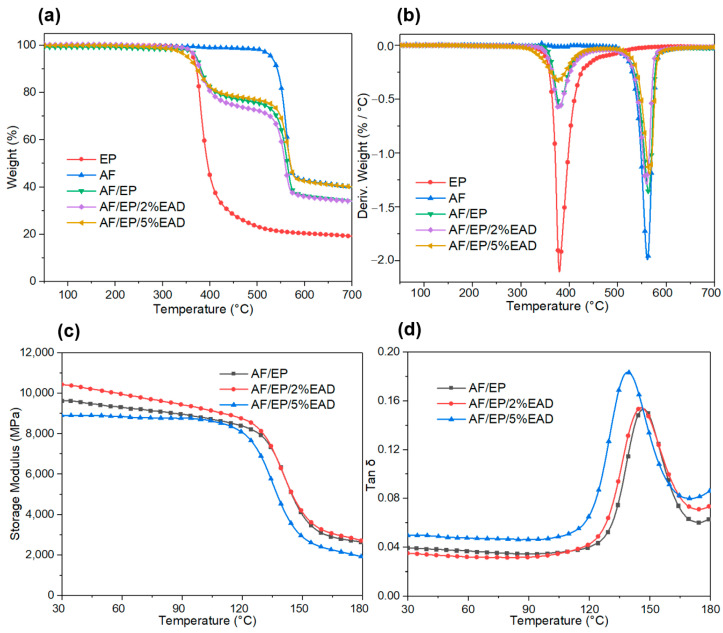
Results from TGA (**a**,**b**) and DMA (**c**,**d**).

**Figure 3 materials-17-04028-f003:**
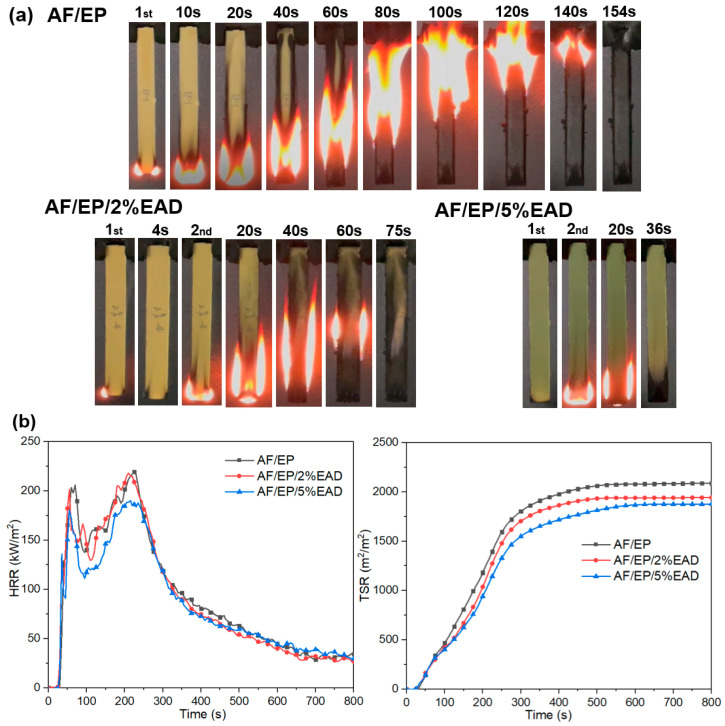
Screenshots of AF/EP composites during UL-94 vertical test (**a**) and data from CCT (**b**).

**Figure 4 materials-17-04028-f004:**
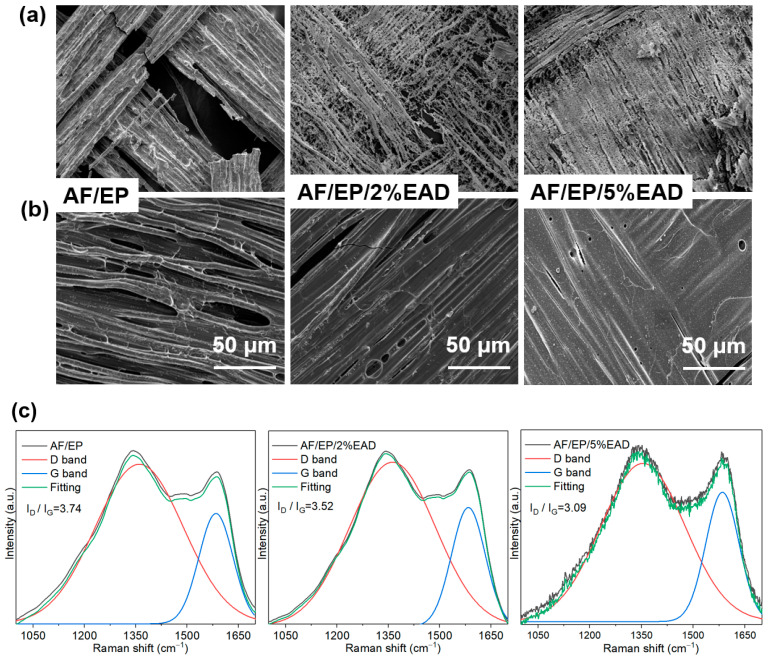
Residue analysis of AF/EP composites after CCT: (**a**) external surface; (**b**) internal surface; (**c**) Raman spectra.

**Figure 5 materials-17-04028-f005:**
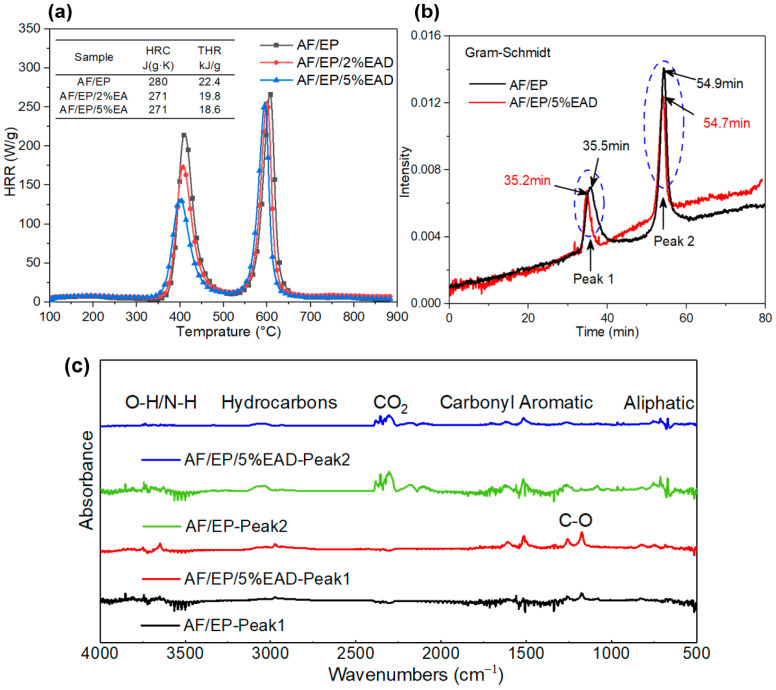
Analysis of gas products from AF/EP composites: (**a**) data from PCFC; (**b**) the evolution of total gas products during the whole degradation process; (**c**) FTIR spectra at the temperatures corresponding to two maximum degradation rates.

**Figure 6 materials-17-04028-f006:**
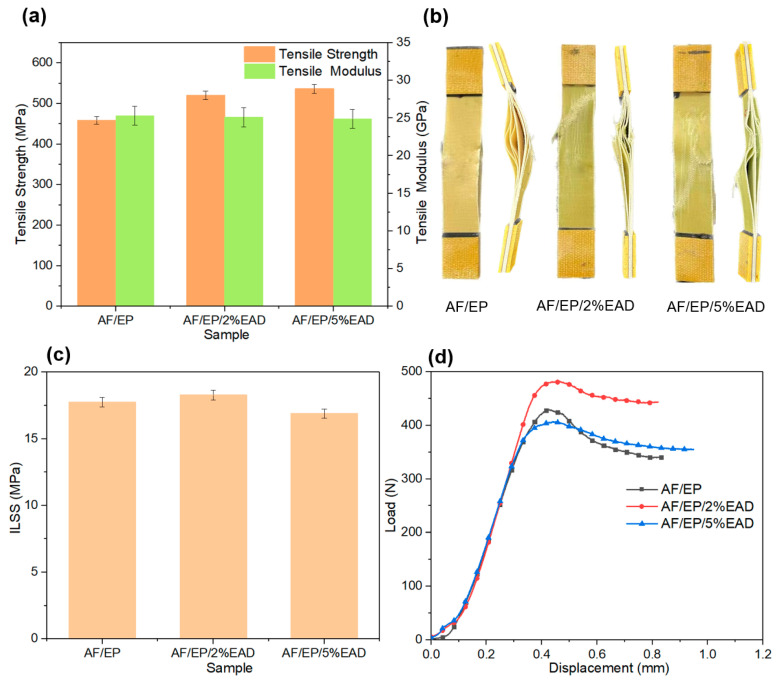
Mechanical data of AF/EP composites from tensile test (**a**,**b**) and ILSS test (**c**,**d**).

**Figure 7 materials-17-04028-f007:**
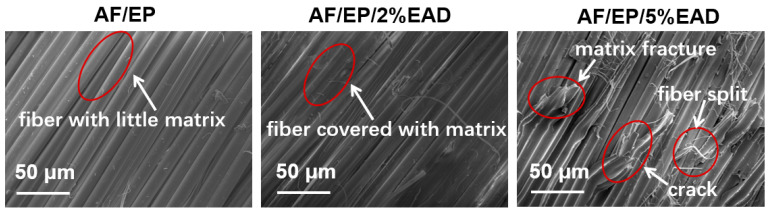
SEM images of AF/EP laminates after tensile failure.

**Figure 8 materials-17-04028-f008:**
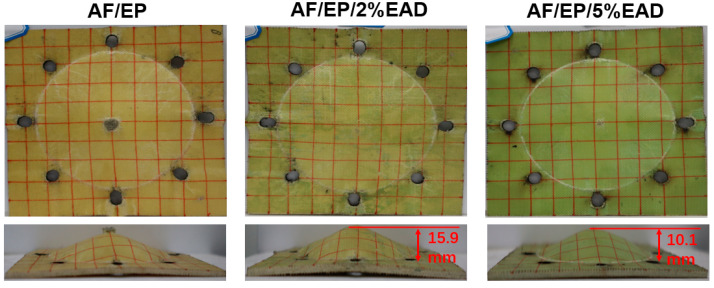
The back face deformation of AF/EP laminates after ballistic test at the same Vi value of 370 ± 1 m/s.

**Table 1 materials-17-04028-t001:** Data of AF/EP composites from CCT.

Sample	TTI (s)	PHRR (kW/m^2^)	MARHE (kW/m^2^)	THR (MJ/m^2^)	av-EHC (MJ/kg)	TSR (m^2^/m^2^)	TSP (m^2^)	Residue (%)
AF/EP	30 ± 1	226 ± 6	152 ± 5	74 ± 1.6	23.1 ± 0.3	2084 ± 80	18.4 ± 1.0	25.3 ± 1.0
AF/EP/2%EAD	26 ± 1	222 ± 5	147 ± 3	71 ± 0.5	22.9 ± 0.2	1938 ± 70	17.1 ± 0.9	28.4 ± 0.6
AF/EP/5%EAD	25 ± 1	201 ± 10	134 ± 7	68 ± 0.5	22.6 ± 0.3	1878 ± 22	16.5 ± 0.2	31.1 ± 1.0

**Table 2 materials-17-04028-t002:** Ballistic impact data of AF/EP composites.

Sample	V_50_ (m/s)	Areal Density (kg/ m^2^)	Specific V_50_ (m^3^/s/kg)	EA (J)	SEA (J·m^2^/kg)
AF/EP	350.5	2.46	142.6	129.3	52.5
AF/EP/2%EAD	372.2	2.58	144.2	145.3	56.3
AF/EP/5%EAD	384.1	2.67	142.7	152.4	57.1

## Data Availability

Data are contained within the article and Appendix A.

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
