# Peer review of "Modified Epoxy Resin on the Burning Behavior and Mechanical Properties of Aramid Fiber Composite"

_materials, 2024, doi:10.3390/ma17164028_

Round 1

Reviewer 1 Report

Comments and Suggestions for Authors

There is more detail in the manuscript and places where some of the claims need support. Especially in the conclusion part, it is important to summarize the results better and support the claims with evidence. Please consider the following comments to improve the manuscript.

I would also like to point out that I really like the visual layout and shapes.

(1) after Figure 5 title, will the the statement “Interlaminar shear property.” be the title of the parts after this sentence?

(2) After Figure 2, there is another Figure 2.

(3) The authors say "the incorporation of EAD solved the inconsistent issue between flame retardancy and mechanical properties"

It claims that EAD solves all problems, but no clear evidence is presented in the conclusion of the article.

(4) please add the following studies to introduction to improve introduction and discussion sections of the manuscript.

https://doi.org/10.3390/jcs7070266

https://doi.org/10.3390/app14093800

(5) The phrase “improved interfacial bonding” requires more detail as to what bonding improvement is achieved. It is not stated by what mechanisms this improvement was achieved.

(6) The authors say "the mass loss of the AF/EP composites before 500 were around 30%"

This statement is a bit vague. It needs to be stated more clearly in which temperature range there is 30% mass loss.

(7) The authors say "the wick effect of aramid fiber and flammability of epoxy resin led to a strong burning behavior"

It requires a more detailed explanation of the cause and effect relationship.

(8) Authors say "the flame retardant can enhance the bonding strength of the matrix-enriched region"

It requires further explanation of how this improvement occurs mechanistically.

(9) It is stated that "the value of T5wt% and Tg slightly decreased by 10 ℃, and 6 ℃, respectively"

It should provide more information as to why this decrease is important.

Author Response

Thank you very much for taking the time to review this manuscript. All the comments have been answered point-by-point. Please find the detailed responses in the attached file and the corresponding revisions were highlighted in yellow background in the revised files.

Reviewer 2 Report

Comments and Suggestions for Authors

Aramid fiber/epoxy resin (AF/EP) composite has been heavily used as impact protection material due to the excellent mechanical properties and lightweight merits. Meanwhile, it is also necessary to concern the flammability of matrix resin and wick effect of aramid fiber, which would cause fire risk in harsh environment.

In this work, a multifunctional flame-retardant modifier (EAD) was incorporated into the AF/EP system to improve the flame retardation.  Dues to the good compatibility and interfacial adhesion, the incorporation of EAD solved the inconsistent issue between flame retardancy and mechanical properties of the composite.  The paper could be considered for publication after revision.

-The authors should compare properties of the new composite with those of other impact protection materials, which are described in literature or which are commercially available.

- How authors would explain that addition of 5 wt% EAD made AF/EP composite with the best properties ?

- It should be explained how AF was cleaned by ethanol ?

- The percentage of flame retardant was set as 2 wt% and 5 wt% in total amount of epoxy resin. Why other amounts were not tested ? Maybe 7-10 wt% of EAD would give better results ?

- Line 158 : ??? The results are presented in  ??? Błąd! Nie można odnaleźć źró- 158 dÅ‚a odwoÅ‚ania ???.

- Chemical structures of AF and EP should be demonstrated.

- Advantages and disadvantages of the new composite should be described clearly in conclusions.

Author Response

(The authors gave the same response as above.)

Reviewer 3 Report

Comments and Suggestions for Authors

The work presents interesting results, but some points are unclear.

In the abstract, the main findings of the research should be presented. Despite stating what was improved, the authors do not present the main results of their research.

The authors present a superficial approach to AF and EP in their introduction. The state-of-the-art on this topic is superficial. What makes your work unique compared to other similar works in the literature? There are several works that combine AF and EP with fire retardants to obtain composites, what makes yours new?

In terms of mass or volume, what is the ratio of AF to EP? I believe that this information is very important. As a result of the authors' choice of AF/EP ratio, what is the motivation for their choice? Is there a reason why other AF/EP concentrations were not used?

Why only 2 and 5 mass% flame retardants were selected by the authors is unclear. Is there a reason why other concentrations were not considered?

The phrase "Błąd! Nie można odnaleźć źró dÅ‚a odwoÅ‚ania" found in line 158 of page 4 is not clear. What does it mean?

In Figure 2, what are the main events that contribute to mass loss in AF and EP?

The TG and DMA analyses should be separated into separate figures. It is strange that the authors separate the results of TGA and DMA into topics, but combine the figures together.

It is unclear why the flame retardant dispersion of 2 and 5 mass% affects the intensity of the DTGA Tm1 and Tm2 peaks!

An in-depth study of DMA analysis is necessary. There is a superficial explanation for the behavior of the curves.

The results of the tensile mechanical analysis are somewhat unusual, since the tensile strength has increased, but the modulus strength has remained almost unchanged. How can these behaviors be explained? From the explanation, it is not clear.

A comparative analysis of the results of the study should be conducted by the authors with those of other studies.

Based on your findings, do you believe that the obtained compounds are suitable for the intended uses?

The manuscript should be thoroughly reviewed to ensure that there are no errors or typos.

Author Response

(The authors gave the same response as above.)

Round 2

Reviewer 2 Report

Comments and Suggestions for Authors

Accept in present form

Reviewer 3 Report

Comments and Suggestions for Authors

The authors responded to all of my inquiries and made improvements to the manuscript. In this regard, I am in favor of the article's acceptance in its current form.